# Haplotype Analysis of *GJB2* Mutations: Founder Effect or Mutational Hot Spot?

**DOI:** 10.3390/genes11030250

**Published:** 2020-02-27

**Authors:** Jun Shinagawa, Hideaki Moteki, Shin-ya Nishio, Yoshihiro Noguchi, Shin-ichi Usami

**Affiliations:** 1Department of Otorhinolaryngology, Shinshu University School of Medicine, 3-1-1 Asahi, Matsumoto, Nagano 390-8621, Japan; junjun0166@shinshu-u.ac.jp (J.S.); moteki@shinshu-u.ac.jp (H.M.); 2Department of Hearing Implant Sciences, Shinshu University School of Medicine, 3-1-1 Asahi, Matsumoto, Nagano 390-8621, Japan; nishio@shinshu-u.ac.jp (S.-y.N.); noguchiy@shinshu-u.ac.jp (Y.N.); 3Department of Otorhinolaryngology, International University of Health and Welfare, Mita Hospital, 1-4-3 Mita, Minato-ku, Tokyo 108-8329, Japan

**Keywords:** *GJB2*, congenital hearing loss, haplotype analysis, founder effect, mutational hot spot, genetic clock

## Abstract

The *GJB2* gene is the most frequent cause of congenital or early onset hearing loss worldwide. In this study, we investigated the haplotypes of six *GJB2* mutations frequently observed in Japanese hearing loss patients (i.e., c.235delC, p.V37I, p.[G45E; Y136X], p.R143W, c.176_191del, and c.299_300delAT) and analyzed whether the recurring mechanisms for each mutation are due to founder effects or mutational hot spots. Furthermore, regarding the mutations considered to be caused by founder effects, we also calculated the age at which each mutation occurred using the principle of genetic clock analysis. As a result, all six mutations were observed in a specific haplotype and were estimated to derive from founder effects. Our haplotype data together with their distribution patterns indicated that p.R143W and p.V37I may have occurred as multiple events, and suggested that both a founder effect and hot spot may be involved in some mutations. With regard to the founders’ age of frequent *GJB2* mutations, each mutation may have occurred at a different time, with the oldest, p.V37I, considered to have occurred around 14,500 years ago, and the most recent, c.176_191del, considered to have occurred around 4000 years ago.

## 1. Introduction

Congenital hearing loss affects approximately one in 500–1000 infants in developed countries, and genetic causes account for at least 50% of all childhood onset non-syndromic sensorineural hearing loss [1]. Currently, it is estimated that there are more than 100 causative genes related to non-syndromic hereditary hearing loss [2], with the most frequent deafness-associated gene worldwide being the *GJB2* gene. Hearing loss caused by *GJB2* gene mutations is divided into autosomal recessive inheritance (DFNB1A) and autosomal dominant inheritance (DFNA3A), but most cases of *GJB2*-associated hearing loss are autosomal recessive inheritance. The allele frequency of *GJB2* gene mutations in the normal Japanese population is approximately 2% [3]. Regarding *GJB2* mutations, recurrent mutations are known to differ among ethnic groups. For example, the c.35delG mutation is commonly observed in European, American, North African, and Middle Eastern populations, but this mutation is rarely observed in the Japanese population, whereas the c.235delC mutation is commonly observed in the Japanese population, but this mutation is relatively rare in European and American populations [4]. Therefore, it is important to clarify the mutation spectrum in each population. In particular, the identification of recurrent mutations is crucial for molecular diagnosis to allow decision-making with regard to the appropriate intervention. Generally, recurrent genetic mutations occur via two mechanisms: one is a founder effect and the other is a mutational hot spot. Interestingly, there are great variations in the prevalence of patients with the *GJB2* mutation in each population, suggesting that the allele frequency in the population, which reflects a founder effect, strongly affects the status of the *GJB2* gene in the deafness population. 

Indeed, the c.35delG mutation in the *GJB2* gene has been proven to be due to a founder effect by haplotype analysis using single nucleotide polymorphisms (SNPs) [5]. Recently, not only *GJB2*, but mutations in various other genes have been extensively studied and the establishment of these recurrent mutations due to a founder effect or a mutational hot spot clarified [6,7,8,9].

In this study, six mutations in the *GJB2* gene commonly observed in the Japanese population were analyzed by SNP-based haplotype analysis to estimate whether the recurring mechanisms for each mutation were due to a founder effect or a mutational hot spot. Also, as founder effects have received special interest in terms of human migration, to address questions about the origin of the founder effect, we also calculated the age at which each mutation considered to be established by a founder effect in this study using the principle of genetic clock analysis [10].

## 2. Materials and Methods

### 2.1. Subjects

We enrolled 7408 sensorineural hearing loss patients, and extracted about 20 patients with each homozygous *GJB2* mutation frequently identified in the Japanese population (i.e., c.235delC, p.V37I (c.109G>A), p.[G45E; Y136X] (c.[134G>A; 408C>A]), p.R143W (c.427C>T), c.176_191del, and c.299_300delAT) (Figure 1) [4]. For the mutations with fewer patients, we also included patients with compound heterozygous mutations, including c.235delC. By using these patients, it was possible to estimate the haplotype for each mutation by eliminating the c.235delC haplotype. This study was conducted in accordance with the Declaration of Helsinki, and the protocol was approved by the Ethics Committee of Shinshu University School of Medicine No. 387—4 September 2012, and No. 576—2 May 2017.

### 2.2. Mutation Analysis

Amplicon libraries were prepared using an Ion AmpliSeq™ Custom Panel (ThermoFisher Scientific, MA, USA), in accordance with the manufacturer’s instructions, for 68 genes reported to cause non-syndromic hereditary hearing loss. After preparation, emulsion PCR and sequencing were performed according to the manufacturer’s instructions. The detailed protocol has been described elsewhere [11]. MPS was performed with an Ion Torrent Personal Genome Machine (PGM) system using an Ion PGM™ 200 Sequencing Kit (ThermoFisher Scientific) and an Ion 318™ Chip (Life Technologies). The sequence data were mapped against the human genome sequence (build GRCh37/hg19) with a Torrent Mapping Alignment Program. After sequence mapping, the DNA variant regions were piled up with Torrent Variant Caller plug-in software (ThermoFisher Scientific). After variant detection, their effects were analyzed using ANNOVAR software [12]. After annotation, we selected the patients with biallelic pathogenic *GJB2* mutations which were reported previously. Direct sequencing was utilized to confirm the selected patients.

### 2.3. SNP Analysis

Haplotypes within the 2 Mbp region surrounding the position of the most frequent mutation (c.235delC) were characterized using a set of 23 SNPs (11 sites upstream and 12 sites downstream). The most representative Tag SNPs were selected at approximately 100,000 bp intervals. For selecting each SNP, we referred to the allelic frequencies in the Integrative Japanese Genome Variation Database (cf. https://ijgvd.megabank.tohoku.ac.jp/). For the Tag SNPs with extremely biased allelic frequencies (e.g., C: 97%, T:3%), we chose other Tag SNPs near this interval (Figure 2). Haplotype analysis was performed using the direct sequencing method.

### 2.4. Statistical Analysis

The linkage disequilibrium range was examined by comparing the allele frequency of each SNP for the hearing loss patients analyzed in this study to the allele frequency in the 3.5KJPN population in the Integrative Japanese Genome Variation Database. Briefly, the allele frequency obtained in this study and the allele frequency in the 3.5KJPN population were examined by using the X^2^ test, and those with a significant difference were regarded as SNPs with linkage disequilibrium. To estimate the linkage disequilibrium region, we used the following criteria; 1) for two continuous SNPs showing *p* > 0.05, this region was not considered to show linkage disequilibrium, and 2) SNPs with allele frequencies ranging from 0.45–0.55 were not included in the linkage disequilibrium.

### 2.5. Estimation of the Occurrence of Each Recurrent Mutation

The estimation of the age at which each mutation occurred was calculated using the equation
$$P_{mO}-P_{O}=\left(1-P_{O}\right)e^{-ct}$$
where *P_mO_* is the frequency of the marker allele *O* on all chromosomes bearing the mutation M, *P_O_* is the frequency of the marker allele *O* on all chromosomes in the normal population, *c* is the recombination rate per generation (we used the value: one recombination for every 1,000,000 bp per generation [13]), and *t* is the number of generations [10].

## 3. Results

A total of 263 patients with homozygous *GJB2* mutations were identified among the 7408 patients (c.235delC: 192 cases, p.V37I: 39, p.[G45E; Y136X]: 17, p.R143W: 6, c.176_191del: 2, and c.299_300delAT: 7). For c.235delC, p.V37I and p.[G45E; Y136X], we performed haplotype analysis of the patients with homozygous mutations. However, for p.R143W, c.176_191del and c.299_300delAT mutations, there were only a small number of patients with homozygous mutations. Thus, we performed haplotype analysis for the patients with compound heterozygous mutations, including c.235delC. By using these compound heterozygous cases, we could determine the haplotype for each mutation by eliminating the estimated haplotype for c.235delC.

The detailed SNP genotypes of each patient for each *GJB2* mutation are shown in Appendix A and the summarized genotypes are shown in Table 1. Patients with each mutation, other than p.V37I, showed a conserved haplotype and the same genotypes were observed in the region close to the target mutations. This might be the result of linkage disequilibrium. For each mutation, the linkage disequilibrium range was 265,063 bp for patients with c.235delC mutations, 665,166 bp for patients with p.[G45E; Y136X] mutations, 229,345 bp for patients with p.R143W mutations, 301,883 bp for patients with c.176_191del mutations, and 301,883 bp for patients with c.299_300delAT mutations (Table 1). We ignored the highly biased SNPs in the 3.5KJPN population to estimate the region of linkage disequilibrium.

On the other hand, in patients with p.V37I, no linkage disequilibrium was observed in our SNP analysis and SNPs very close to the p.V37I mutation (5’SNP1) also differed among the patients. However, if the patients were divided by the 5′SNP1 residue (i.e., the C group or T group), common haplotypes could be confirmed in the 3’SNPs. The C residue group showed a G residue in 3’SNP2, G in 3’SNP4, and G in 3’SNP6, while the T residue group showed different haplotypes with A in 3’SNP2, A in 3’SNP4, and A in 3’SNP6. The linkage disequilibrium range for the C residue group was 80,923 bp, whereas that for the T residue group was 301,883 bp.

Thus, we concluded that all six mutations occurred due to founder effects, and we next estimated the year at which each mutation occurred by using the length of the linkage disequilibrium and the equation described in the Methods section. If we assume that one generation is 25 years, it was predicted that c.235delC occurred around 6500 years ago, p.[G45E; Y136X] occurred around 6000 years ago, p.R143W occurred around 6500 years ago, c.176_191del occurred around 4000 years ago, and c.299_300delAT occurred around 7700 years ago. Further, it was predicted that the founder of the C residue group in 5’SNP1 for p.V37I occurred around 14,500 years ago and the founder of the T residue group in 5’SNP1 for p.V37I occurred around 5000 years ago.

## 4. Discussion

The present results indicated that all six mutations frequently observed in the Japanese population seem to be founder mutations.

The geographic regional distribution of each *GJB2* mutation can also be indicative of whether the mutation is a founder mutation or hot spot mutation. If the mutations are clearly found in only a limited ethnic population, it is possible to predict those mutations are due to a founder effect; conversely, if the mutations appear uniformly all over the world or in many ethnic populations, then the mutations can be considered to be due to a mutational hot spot.

Regarding the c.235delC mutation, a series of previous studies based on haplotype analysis concluded that this mutation was caused by a founder effect [14,15,16], with the same result obtained in this study. Our previous haplotype analysis using six SNPs in 16 homozygous and 92 heterozygous c.235delC patients and in 90 controls without the 235delC mutation, indicated that the c.235delC mutation is derived from a common ancestor because we found common alleles on two SNPs near the c.235delC [10]. Similarly, Yan et al. performed haplotype analysis using seven SNPs near the c.235delC for 45 unrelated patients carrying the c.235delC mutation and found common alleles on their SNPs, so they also concluded that the c.235delC mutation is caused by a founder effect [16]. 

Based on our review of the *GJB2* mutation spectrum, each population around the world has a specific spectrum [4]. According to the results, the c.235delC mutation is frequently observed in countries in East and Central Asia, such as Japan, Korean, China, Mongolia, and Thailand, whereas it is rarely observed in other countries. The uneven distribution of this mutation also suggests that it was caused by a founder effect.

Based on the present data, the occurrence of the c.235delC mutation is estimated at around 6500 years ago. Yan et al. reported that the c.235delC mutation may have occurred about 11,500 years ago [16]. The reason for the differences in the estimated time between two studies may be due to the different SNPs used and the number of subjects.

The c.299_300delAT mutation is reported to be found in the Japanese, Chinese, Korean, Mongolian, Australian, Turkish, Romanian, and American populations and is especially frequently observed in East Asian countries [4]. Current haplotype analysis indicated that this mutation is estimated to have occurred around 7700 years ago. The estimated founders’ age of the c.299_300delAT mutation is compatible with the distribution of the mutation; i.e., the distribution of the c.299_300delAT mutation is relatively wider than those of the c.176_191del mutation and p.[G45E; Y136X] mutation mentioned below.

The c.176_191del mutation is observed in the Japanese, Chinese, American, and Brazilian populations, but is mainly distributed in East Asian countries [4]. We estimated that this mutation occurred around 4000 years ago, a relatively recent event, which is consistent with the limited distribution of this mutation.

The p.[G45E; Y136X] mutation is only observed in the Japanese population [4]. This distribution means that this mutation occurred only in a Japanese ancestor, but this mutation is estimated to have occurred around 6000 year ago based on our results. However, the putative linkage disequilibrium length for this mutation was longer than those of the other mutations, supporting the notion that this mutation occurred more recently. Therefore, the true age at which p.[G45E; Y136X] occurred may be younger than that suggested by our analysis. Further analysis using a larger number of patients may clarify this estimation.

Regarding the p.R143W mutation, Tsukada et al. summarized this mutation as occurring in the Ghanaian, Japanese, Korean, and Argentinean populations at moderate frequencies and also in the Mongolian, Australian, Iranian, Turkish, Estonian, Dutch, Spanish, Swedish, and American populations at low frequencies [4]. Except for the Ghanaian population, the frequency of this mutation in all *GJB2* mutations observed in each population is less than 10%. Otherwise, the frequency of p.R143W in all *GJB2* mutations observed in the Ghanaian population is 90.9%. The wide distribution of this mutation suggests that this mutation occurred as a mutational hot spot or in very old common ancestors. However, the finding that the peripheral region of the p.R143W mutation is conserved in the Japanese population suggests that this mutation was caused by a founder effect and the estimated age of occurrence is 6500 years ago. This fact suggests that the ancestor of this mutation in Ghana and that in the other countries may be different, and this mutation may have occurred as multiple events. Haplotype analysis of the p.R143W patients in Ghana could clarify this controversy.

Lastly, regarding the p.V37I mutation, according to Dahl et al. [17], haplotype analysis showed that the p.V37I mutation is derived from a founder effect. In their report, the sample size was relatively small with only four subjects examined, so the possibility of selection bias cannot be ruled out. Based on our present results, it would be natural to hypothesize that there were two founders for the p.V37I mutation, although we did not distinguish which haplotype group is the founder for the Australian population reported by Dahl et al. 

Tsukada et al. summarized the p.V37I mutation as frequently occurring in the East Asian, Southeast Asian, and Australian populations [4]. At the same time, this mutation is also observed in the United States, Argentina, and North African countries at low frequencies. This biased distribution of the p.V37I mutation also supports the hypothesis that this mutation occurred due to a founder effect. The wide distribution suggested that this mutation occurred at an older age and probably as multiple events. Indeed, from our results, it was estimated that the founder of the C residue group in 5’SNP1 for p.V37I occurred around 14,500 years ago. In addition, the founder of the T residue group in 5’SNP1 for p.V37I is considered to be relatively new, and may be the type most commonly found in East Asian countries. Haplotype analysis of p.V37I patients in Taiwanese and Chinese patients could clarify this problem.

There are two limitations to the method used for the estimation of the time at which each founder mutation occurred. The first is that we do not know the correct recombination rate of this region. For example, based on the distance from 5’SNP 6 to c.235delC (265,063bp), we used a linear relationship of 1cM, and calculated the recombination rate as 0.00265063 per generation. However, as we do not know the true recombination frequency for this region, this calculation is only a rough estimate. The second limitation is the SNP selection bias. Ideally, SNPs with an allele frequency of 0.5:0.5 are favorable for detecting statistically significant differences. However, for example, the interval of about 250,000 bp between the 5’SNPs 6 and 5’SNPs 7 is open because there were no appropriate SNPs. Since the prediction of the founders’ ages is dependent on the linkage disequilibrium length, more detailed data on the correct recombination rate will solve the problem. 

## 5. Conclusions

This study has shown that frequent mutations in *GJB2* are derived from founder effects rather than hot spots. Also, in some of the frequent mutations (p.R143W and p.V37I) there are potentially multiple origins, indicating that both a founder effect and hot spot may be involved. When considering the fact that there are many ethnically specific mutations, in spite of methodological limitations, this study has shed light on the relative founders’ age of frequent *GJB2* mutations and shown one example of the occurrence of mutational events during human migration.

## Figures and Tables

**Figure 1 genes-11-00250-f001:**
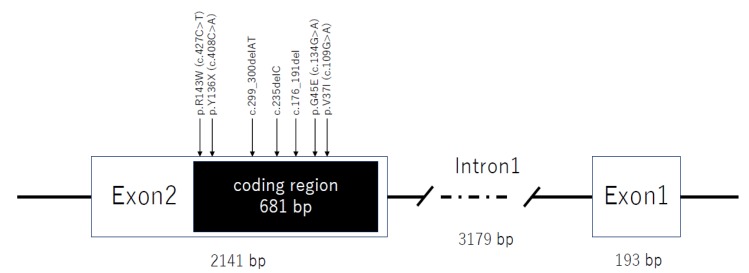
A schematic diagram showing the location of each mutation within the *GJB2* gene. White boxes indicate exons, and the black box in exon 2 indicates the coding region. All six mutations targeted in this study exist in the coding region.

**Figure 2 genes-11-00250-f002:**
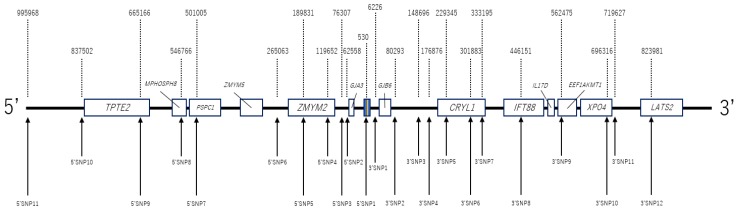
The location of single nucleotide polymorphisms (SNPs). Haplotypes within the 2 Mbp region surrounding the position of the most frequent mutation (c.235delC) were characterized using a set of 23 SNPs (11 sites upstream and 12 sites downstream). Tag SNPs were selected at approximately 100,000 bp intervals. For the positions with extremely biased allelic frequencies (e.g., C: 97%, T:3%), SNPs were inevitably set according to the interval. The blue rectangle in the middle indicates the *GJB2* gene. The yellow line in the rectangle indicates c.235delC. Other white rectangles indicate genes around *GJB2*. The numbers above the line indicate the relative distance of each SNP when c.235delC is set to 0. The numbers of the SNPs below the line correspond to the SNP numbering used in this paper.

**Table 1 genes-11-00250-t001:** Results of the haplotype analysis for six frequently observed GJB2 mutations in the Japanese population.

		Distance from c.235delC(bp)	Allele Frequency (Tohoku Medical Megabank Organization)			c.235delC			p.V37I						p.G45E;Y136X			p.R143W			c.176_191del			c.299_300del		
	Marker								5’SNP1_C Group			5’SNP1_T Group														
						Allele Frequency		*p*-Value	Allele Frequency		*p*-Value	Allele Frequency		*p*-Value	Allele Frequency		*p*-Value	Allele Frequency		*p*-Value	Allele Frequency		*p*-Value	Allele Frequency		*p*-Value
5’SNP11	rs9553673	995968	G 5873	C 1231	Total 7104	G:36	C:4	*p* = 0.22	G:15	C:1	*p* = 0.24	G:17	C:3	*p* = 0.78	G:29	C:5	*p* = 0.69	G:17.6	C:10.4	*p* < 0.01	G:21.1	C:0.9	*p* = 0.10	G:15.3	C:11.7	*p* < 0.01
5’SNP10	rs4569114	837502	C 5342	T 1760	Total 7102	C:30	T:10	*p* = 0.97	C:10	T:6	*p* = 0.24	C:14	T:6	*p* = 0.59	C:24	T:10	*p* = 0.53	C:21.25	T:6.75	*p* = 0.93	C:10.25	T:11.75	*p* < 0.01	C:16	T:11	*p* = 0.06
5’SNP9	rs4769700	665166	A 5228	G 1874	Total 7102	A:28	G:12	*p* = 0.61	A:11	G:5	*p* = 0.66	A:12	G:8	*p* = 0.17	A:20	G:14	*p* = 0.05	A:22.5	G:5.5	*p* = 0.42	A:10	G:12	*p* < 0.01	A:16.1	G:10.9	*p* = 0.10
5’SNP8	rs8000138	546766	A 4624	G 2482	Total 7106	A:29	G:11	*p* = 0.33	A:15	G:1	*p* = 0.02	A:17	G:3	*p* = 0.06	A:25	G:9	*p* = 0.30	A:15.65	G:12.35	*p* = 0.31	A:15.2	G:6.8	*p* = 0.69	A:12.2	G:14.8	*p* = 0.03
5’SNP7	rs3742148	501005	A 4946	G 2158	Total 7104	A:27	G:13	*p* = 0.77	A:13	G:3	*p* = 0.31	A:15	G:5	*p* < 0.01	A:14	G:18	*p* < 0.01	A:18.25	G:9.75	*p* = 0.61	A:13.6	G:8.4	*p* < 0.01	A:17.925	G:9.075	*p* = 0.72
5’SNP6	rs4769920	265063	A 3556	G 3550	Total 7106	A:30	G:10	*p* < 0.01	A:9	G:7	*p* = 0.62	A:12	G:8	*p* = 0.37	A:9	G:25	*p* < 0.01	A:18	G:10	*p* = 0.13	A:13.75	G:8.25	*p* = 0.24	A:9.5	G:17.5	*p* = 0.12
5’SNP5	rs9508995	189831	G 4610	C 2496	Total 7106	G:32	C:8	*p* = 0.05	G:12	C:4	*p* = 0.40	G:13	C:7	*p* = 0.99	G:9	C:25	*p* = 0.01	G:21	C:7	*p* = 0.26	G:14.4	C:7.6	*p* = 0.95	G:18.6	C:8.4	*p* = 0.66
5’SNP4	rs9509023	119652	T 5517	C 1589	Total 7106	T:35	C:5	*p* = 0.14	T:14	C:2	*p* = 0.34	T:17	C:3	*p* = 0.43	T:14	C:20	p<0.01	T:22.625	C:5.375	*p* = 0.69	T:15.625	C:6.375	*p* = 0.46	T:21.375	C:5.625	*p* = 0.85
5’SNP3	rs2031282	76307	G 6863	A 215	Total 7078	G:40	A:0	*p* = 0.26	G:16	A:0	*p* = 0.48	G:19	A:1	*p* = 0.61	G:30	A:2	*p* = 0.29	G:27	A:1	*p* = 0.87	G:22	A:0	*p* = 0.41	G:25	A:2	*p* = 0.19
5’SNP2	rs747931	62558	T 6271	C 783	Total 7054	T:39	C:1	*p* = 0.08	C:1	T:15	*p* = 0.54	C:0	T:20	*p* = 0.11	T:34	C:0	*p* = 0.04	T:27.025	C:0.975	*p* = 0.20	T:20.05	C:1.95	*p* = 0.74	T:27	C:0	*p* = 0.07
5’SNP1	rs3751385	530	C 3975	T 3133	Total 7108	C:40	T:0	*p* < 0.01	C:15	T:1	*p* < 0.01	C:4	T:16	*p* < 0.01	C:34	T:0	*p* < 0.01	C:28	T:0	*p* < 0.01	C:22	T:0	*p* < 0.01	C:27	T:0	*p* < 0.01
		0																								
3’SNP1	rs5030702	6226				A:40	C:0	*p* = 0.45	A:16	C:0	*p* = 0.60	A:20	C:0	*p* = 0.56	A:34	C:0	*p* = 0.48	A:28	C:0	*p* = 0.51	A:22	C:0	*p* = 0.55	A:27	C:0	*p* = 0.52
3’SNP2	rs7324573	80923	A 3451	G 3439	Total 6890	A:39	G:1	*p* < 0.01	A:3	G:13	*p* = 0.01	A:19	G:1	*p* < 0.01	A:30	G:4	*p* < 0.01	A:5.325	G:22.675	*p* < 0.01	A:21	G:1	*p* < 0.01	A:12	G:15	*p* = 0.56
3’SNP3	rs9579842	148696	C 4237	G 2871	Total 7108	C:14	G:26	*p* < 0.01	C:13	G:3	*p* = 0.08	C:15	G:5	*p* = 0.16	C:12	G:22	*p* < 0.01	C:3.25	G:24.75	*p* < 0.01	C:5.25	G:16.75	*p* < 0.01	C:7.2	G:19.8	*p* < 0.01
3’SNP4	rs9509177	176876	G 4273	A 2733	Total 7006	G:33	A:7	*p* < 0.01	G:13	A:3	*p* = 0.10	G:4	A:16	*p* < 0.01	G:25	A:9	*p* = 0.14	G:19.7	A:8.3	*p* = 0.31	G:20.35	A:1.65	*p* < 0.01	G:10.4	A:16.6	*p* = 0.02
3’SNP5	rs7332444	229345	C 4848	T 2252	Total 7100	C:31	T:9	*p* = 0.21	C:11	T:5	*p* = 0.97	C:17	T:3	*p* = 0.11	C:25	T:9	*p* = 0.51	C:24	T:4	*p* = 0.05	C:4.7	T:17.3	*p* < 0.01	C:17.9	T:9.1	*p* = 0.83
3’SNP6	rs2872488	301883	G 3938	A 3166	Total 7104	G:19	A:21	*p* = 0.32	G:14	A:2	*p* = 0.01	G:5	A:15	*p* = 0.01	G:18	A:16	*p* = 0.77	G:14.675	A:13.325	*p* = 0.75	G:4.675	A:17.325	*p* < 0.01	G:9.05	A:17.95	*p* = 0.02
3’SNP7	rs9509266	333195	C 4710	G 2374	Total 7084	C:30	G:10	*p* = 0.26	C:12	G:4	*p* = 0.47	C:10	G:10	*p* = 0.12	C:25	G:9	*p* = 0.39	C:21.25	G:6.75	*p* = 0.29	C:18	G:4	*p* = 0.13	C:20.75	G:6.25	*p* = 0.26
3’SNP8	rs9509311	446151	A 4865	G 2243	Total 7108	A:11	G:21	*p* < 0.01	A:15	G:1	*p* = 0.03	A:18	G:2	*p* = 0.04	A:21	G:13	*p* = 0.40	A:18.8	G:9.2	*p* = 0.88	A:19	G:3	*p* = 0.07	A:13.6	G:13.4	*p* = 0.04
3’SNP9	rs9506549	562475	T 5018	C 2084	Total 7102	T:17	C:17	*p* < 0.01	T:14	C:2	*p* = 0.14	T:17	C:3	*p* = 0.16	T:12	C:10	*p* = 0.10	T:15.15	C:12.85	*p* = 0.06	T:19.25	C:2.75	*p* = 0.08	T:12.25	C:14.75	*p* < 0.01
3’SNP10	rs7330520	696316	A 5623	G 1481	Total 7104	A:32	G:6	*p* = 0.44	A:14	G:2	*p* = 0.41	A:19	G:1	*p* = 0.08	A:26	G:8	*p* = 0.70	A:16.63	G:11.37	*p* = 0.01	A:18.63	G:3.37	*p* = 0.52	A:16.263	G:10.737	*p* = 0.02
3’SNP11	rs9579970	719627	T 4448	C 2658	Total 7106	T:22	C:10	*p* = 0.47	T:15	C:1	*p* = 0.01	T:9	C:11	*p* = 0.10	T:26	C:8	*p* = 0.10	T:21.9375	C:6.0625	*p* = 0.09	T:15.875	C:6.125	*p* = 0.36	T:15.875	C:11.125	*p* = 0.68
3’SNP12	rs2050576	823981	C 4042	T 3060	Total 7102	C:19	T:19	*p* = 0.39	C:11	T:5	*p* = 0.34	C:12	T:8	*p* = 0.78	C:23	T:11	*p* = 0.21	C:13.5	T:14.5	*p* = 0.35	C:14	T:8	*p* = 0.53	C:14	T:13	*p* = 0.60

The blue lines show the linkage disequilibrium range. The yellow boxes show that there is a significant difference between the allele frequency obtained in this study and the allele frequency in the Integrative Japanese Genome Variation Database as assessed by the X^2^ test. The gray boxes show that there is no significant difference between the allele frequency obtained in this study and the allele frequency in the Integrative Japanese Genome Variation Database as assessed by X^2^ test. The green boxes show that there is no significant difference due to the originally biased allele frequency in the Integrative Japanese Genome Variation Database. The red boxes show that the allele frequency obtained this study is from 0.45 to 0.55.

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
