# Peer review of "Haplotype Analysis of GJB2 Mutations: Founder Effect or Mutational Hot Spot?"

_genes, 2020, doi:10.3390/genes11030250_

Round 1

Reviewer 1 Report

GBJ2 is an important deafness gene.  The authors do not explain in their introduction current theories of what this may do.  The introduction does not explain why their study is important - and as it stands, presents as a study that can be done as the technology is available.  Whilst I do accept the focus of their study is on the origin of the mutations (founder or hotspot), it would be good to have some discussion concerning what - if anything - the mutations they study have to say about the gene/or its dysregulation.  This will help prepare the argument as to why their study is important.  Ultimately this may help the authors why is it of interest to know the origin of a mutation? 

Line 45-6.  Some references would be pertinent here.

Figure 1 can be improved.  There is no figure legend.  What is the rectangle in the centre - the gene? the c235 deletion? The numbers above correspond to chromosome 13 position?  The SNP numbers above?  The numbers below correspond to the SNP numbers used in this study?  Where is this in relation to the genome sequence/and location 13q12.11 - and neighbouring genes?  Size markers along the 2Mbp region?  There needs more work on this figure and of course a figure legend.

Line 146-150.  The exons of the GJB2 gene occupy a relatively small region of the chromosome.  Therefore it would be very helpful to have a small figure, that shows the position of these mutations.   Indeed, it would be even more helpful to see the location of mutations from other populations.  Essentially some form of more clearly demonstrating the point the authors are making would improve the manuscript.  This would also frame the following discussion on individual alleles.  I am still unclear if these are mutations in the protein or outside and I did assume that all mutations are in the encoded region of GJB2?   Note added on third reading - I think there is some confusion.  Reading the manuscript twice I took your use of regional to mean in the chromosomal region, whereas I think you actually mean geographically regional!  I would rewrite this so as others do not suffer the same confusion as myself!   

The manuscript is well written and competing/contrasting results/calculations are addressed.  My main negative comment is that nothing is made of any functional inferences, if any, can be made from the mutations studied as this would add to the importance of such a study.  I would expect the authors will claim that is not the target of this paper, but it will be for many readers who look at the manuscript.  Perhaps also to round up the paper is whether mutations from different geographical regions, although independent, target similar areas of GJB2 helping to identify critical regions of the protein?

Reviewer 2 Report

The study by Shinagawa et al. presents interesting data on haplotype analysis of GJB2 mutations frequently observed in Japanese haring loss patients. Based on the results they conclude that the mutations derive from founder effects and they attempt to estimate the age of their occurrence.

  1. In the manuscript the authors should explain why it is important to differentiate between a hot spot mutation and a founder effect. It might be not clear for a reader why the study has been conducted. Does it have any practical significance?
  2. Please add references to the sentence ending in line 46. What are the other genes?
  3. The variant nomenclature should b unified. Some mutations are given at the “c.” cDNA level and other at the “p.” protein level.
  4. The fragment in lines 83-86 is almost identical to the text written above (lines 74-79).
  5. In the Figure 1 it is not clear what the numbers in the diagram mean. What does the blue rectangle represent?
  6. In Table parts of variant names and parts of variant number have been move to a second line, which makes the table difficult to follow. Please improve the formatting of Table 1.
